Insights on the evolution of Coronavirinae in general, and SARS-CoV-2 in particular, through innovative biocomputational resources

Dos Santos Daniel Andrés dadossantos@csnat.unt.edu.ar 1 2
Reynaga María Celina celinareynaga@gmail.com 2
González Juan Cruz 2
Fontanarrosa Gabriela 2
Gultemirian María de Lourdes 2 3
Novillo Agustina 2
Abdala Virginia 2 4
1 Cátedra de Bioestadística, Facultad de Ciencias Naturales e Instituto Miguel Lillo, Universidad Nacional de Tucumán , San Miguel de Tucumán , Tucumán , Argentina
2 Instituto de Biodiversidad Neotropical, Consejo Nacional de Investigaciones Científicas y Técnicas (CONICET) - Universidad Nacional de Tucumán (UNT) , Yerba Buena , Tucuman , Argentina
3 Cátedra de Química Inorgánica, Facultad de Ciencias Naturales e Instituto Miguel Lillo, Universidad Nacional de Tucumán , San Miguel de Tucumán , Tucumán , Argentina
4 Cátedra de Biología General y Metodología de las Ciencias, Facultadad de Ciencias Naturales e Instituto Miguel Lillo, Universidad Nacional de Tucumán , San Miguel de Tucumán , Tucumán , Argentina
Wiles Siouxsie
Electronic publication date: 2022 Jul 25
Publication date: 2022
Volume: 10
Electronic Location ID: e13700
Received 2022 Feb 14; Accepted 2022 Jun 17
Copyright: ©2022 Dos Santos et al.
Copyright year: 2022
Copyright holder: Dos Santos et al.
License: This is an open access article distributed under the terms of the Creative Commons Attribution License, which permits unrestricted use, distribution, reproduction and adaptation in any medium and for any purpose provided that it is properly attributed. For attribution, the original author(s), title, publication source (PeerJ) and either DOI or URL of the article must be cited.
License URL: https://creativecommons.org/licenses/by/4.0/

Keywords: Coronavirus, Host-virus interaction, Chimerism, Evolutionary constraints, Virology, Zoonotic reservoirs, Viral proteomes, Viral assembly

Funding: CONICET PUE-0099 PIP 652 PICT-2019-04546 PICT 2018-2772 1122015 0100258 CO This work was supported by CONICET’s UE Project: 0099. Celina Reynaga and Juan Cruz González are supported by PIP 652. Gabriela Fontanarrosa is supported by PICT-2019-04546. Virginia Abdala is supported by PICT 2018-2772. Agustina Novillo is supported by PIP CONICET 1122015 0100258 CO. The funders had no role in study design, data collection and analysis, decision to publish, or preparation of the manuscript.

==============================
The structural proteins of coronaviruses portray critical information to address issues of classification, assembly constraints, and evolutionary pathways involving host shifts. We compiled 173 complete protein sequences from isolates belonging to the four genera of the subfamily Coronavirinae. We calculate a single matrix of viral distance as a linear combination of protein distances. The minimum spanning tree (MST) connecting the individuals captures the structure of their similarities. The MST re-capitulates the known phylogeny of Coronovirinae. Hosts were mapped onto the MST and we found a non-trivial concordance between host phylogeny and viral proteomic distance. We also study the chimerism in our dataset through computational simulations. We found evidence that structural units coming from loosely related hosts hardly give rise to feasible chimeras in nature. This work offers a fresh way to analyze features of SARS-CoV-2 and related viruses.

Introduction

Viruses can be considered molecular parasites (Koonin, Dolja & Krupovic, 2015) with an asexual type of reproduction (assisted by cells’ replication mechanisms) in which gene exchanges do not take place. Novel hybrid infectious particles—or chimeras—may be generated when a host cell is infected with at least two viral genomes at the same time (Simon-Loriere & Holmes, 2011). Based on this phenomenon, chimeric viruses, have also been created in laboratories by gain-of-function experiments in order to make, for instance, novel attenuated pan-vaccines (Jochmus et al., 1999; Whitehead et al., 2003; Akhand et al., 2020 among others).

Although viruses cannot be considered as biological entities, they are evolutionary entities; thus, the inquiry about their evolutionary constraints is pertinent. Phenotypic limits in evolutionary entities can be detected by comparing theoretical vs. empirical morphospaces (McGhee, 1999; Eble, 2000). A theoretical morphospace is an n-dimensional space describing and relating phenotypic configurations. It is generated by the systematic variation of the parameters underlying selected construction rules (Raup, 1967; Mitteroecker & Gunz, 2009). An empirical morphospace is the set of entities observed in nature (McGhee, 1999). Viral proteomes can be analyzed under a morphospace framework to delineate a gradient across the domains of existent, plausible, and impossible entities, henceforth the feasibility gradient. Morphospace analytical approaches include tools such as combinatorics, probability, network analysis, among others.

After two years of research and a myriad of publications on SARS-Cov2, the origin of this virus is still a matter not resolved. Moreover, the WHO-led international mission that has begun investigations in China to try to establish the origin of SARS-CoV-2 estimates that could take years to track the zoonotic jumps behind the viral origin (Zarocostas, 2021). The rise of emergent viruses, coming from wild reservoirs can occur as a consequence of increased opportunities for transmission due to perturbations in the ecosystem, independently of changes in their genetic structure (Solé & Elena, 2018). The most closely related virus to SARS-CoV-2 is RaTG13, identified from a Rhinolophus affinis bat. However, the receptor-binding domain (RBD) of SARS-CoV-2 is more similar to its analog to the pangolin-CoV-2020 isolated from Malayan pangolins (Manis javanica) (Segreto & Deigin, 2020). The ambiguous nature of SARS-CoV-2 suggests two main hypotheses regarding its origin: (1) A zoonotic origin with R. affinis as a wild host reservoir, in which mutational changes have occurred in the RBD, conferring the virus the ability to infect humans. This sequence is similar by convergence to the pangolin-CoV-2020; (2) A chimeral origin in which the backbone of the novel virus comes from RaTG13 and the RBD comes from the pangolin-Cov-2020. Interestingly, using an analytic strategy based on shell disorder models, Goh et al. (2020), Goh et al. (2022) offered a renewed view that went against the mainstream on the subject. They measured the percentage of intrinsic disorder of proteins from the viral inner and outer shells, and suggested a silent spreading among humans of a SARS-CoV-2 precursor coming from pangolins. It is, therefore, possible that pangolins play an important role in the ecology and/or evolution of SARS-CoV-2. If this is true, either natural or artificial chimerism becomes a working hypothesis.

Known members of the subfamily Coronavirinae include four genera alpha (α), beta (β), gamma (γ), and delta (δ) (Payne, 2017). Their genome encodes, among others, four main structural proteins essential for viral assembly: envelope (E), membrane (M), nucleocapsid (N), and spike (S) (Yao et al., 2020). E, M, N, and S proteins constitute the basic phenotypic configuration of Coronavirinae that varies between different viral species regarding their specific amino acid sequence. Considering each protein as our building block of the Coronavirinae diversity, we can compose a viral theoretical morphospace as the free combination of those proteins in 4-tuples. Let suppose the next 4-tuple describing the proteinic structure of a given empirical coronavirus x, Ex-Mx-Nx-Sx, and the 4-tuple of another hypothetical empirical coronavirus Ey-My-Ny-Sy from the coronavirus y, and let suppose that hosts in which they were found are Hx and Hy. We hypothesize that any new combination of a theoretical virus (e.g., Ex-My-Ny-Sx) is plausible to the extent that the involved hosts are closely related and viral proteins are similar with regard to their analogs in the observed assembly of empirical viruses (co-occurring proteins).

In this paper, we study the Coronavirinae under an exploration of sequence space. Based on the empirical set of viral protein sequences, we calculate the similarities among them. We simulate a theoretical viral morphospace and propose a scoring system to measure the degree of plausibility of occurrence in nature for every theoretical phenotypical option. We focus on two factors to assess the degree of chimerality for any viral protein assembly: The chemical protein affinities by one side, and the phylogenetic relatedness of the hosts by the other. By chimerality we understand the result of a recombination process as being, for example, the result of a coinfection. Through this approach, (1) we bring the attention to hot topics such as the prediction of unobserved virus lineages, (2) we provide insights to settle debates about the dichotomy artificial-natural emergence of new viruses, and (3) we offer clues to inquire about intermediary hosts in zoonotic jumps.

Materials & Methods

The analysis was based on viral individuals from different vertebrate taxa, belonging to the four genera of Coronavirinae. Therefore, we consulted the NCBI database and utilized a downloaded data table with all the applicable annotations for which complete data of sequences were available for the totality of structural proteins. We end up with a collection of 173 coronaviruses, two of them belonging to unknown genera. Accession numbers in addition to relevant metadata can be found in Table 1. Complete sequences of E, M, N, and S, were aligned using the MUSCLE algorithm (Edgar, 2004) in MEGA X software (Kumar et al., 2018). The number of amino acid substitutions per site was calculated between sequences using the Poisson correction model (Zuckerkandl & Pauling, 1965). The rate variation among sites was modeled with a gamma distribution (shape parameter = 1). These analyses involved 173 amino acid sequences for each class of protein. All positions containing gaps and missing data were eliminated (complete deletion option).

Table 1 Viruses attributes.

List of attributes for the 173 virus sampled from NCBI Virus database. The amino acid sequences of all the four structural proteins (E, M, N and S) are available for these individuals. The ID number is used to identify the isolate in the network of Figure 1. Information includes accession numbers, genome size (nucleotide number), host taxa, and title reported in GenBank for each sequence (very long titles were shortened for clarity).

Node ID	Acc. No.	Size	Host	Genus	GenBank Title	
1	AY278489	29757	NA	β-coronavirus	SARS coronavirus GD01	
2	AY390556	29760	NA	β-coronavirus	SARS coronavirus GZ02	
3	AY572035	29518	Viverridae	β-coronavirus	SARS coronavirus civet010	
4	AY686864	29525	Paradoxurus hermaphroditus	β-coronavirus	SARS coronavirus B039	
5	DQ022305	29728	NA	β-coronavirus	Bat SARS coronavirus HKU3-1	
6	DQ071615	29736	Chiroptera	β-coronavirus	Bat SARS coronavirus Rp3	
7	DQ084200	29711	NA	β-coronavirus	bat SARS coronavirus HKU3-3	
8	DQ412042	29709	Rhinolophus ferrumequinum	β-coronavirus	Bat SARS coronavirus Rf1	
9	DQ412043	29749	Rhinolophus macrotis	β-coronavirus	Bat SARS coronavirus Rm1	
10	DQ648856	29704	NA	β-coronavirus	Bat coronavirus (BtCoV/273/2005)	
11	DQ648857	29741	NA	β-coronavirus	Bat coronavirus (BtCoV/279/2005)	
12	DQ811787	27550	Sus scrofa	α-coronavirus	PRCV ISU-1	
13	EF065505	30286	Chiroptera	β-coronavirus	Bat coronavirus HKU4-1	
14	EF065506	30286	Chiroptera	β-coronavirus	Bat coronavirus HKU4-2	
15	EF065507	30286	Chiroptera	β-coronavirus	Bat coronavirus HKU4-3	
16	EF065508	30316	Chiroptera	β-coronavirus	Bat coronavirus HKU4-4	
17	EF065509	30482	Chiroptera	β-coronavirus	Bat coronavirus HKU5-1	
18	EF065510	30488	Chiroptera	β-coronavirus	Bat coronavirus HKU5-2	
19	EF065511	30488	Chiroptera	β-coronavirus	Bat coronavirus HKU5-3	
20	EF065512	30487	Chiroptera	β-coronavirus	Bat coronavirus HKU5-5	
21	EF065513	29114	Chiroptera	β-coronavirus	Bat coronavirus HKU9-1	
22	EF065514	29107	Chiroptera	β-coronavirus	Bat coronavirus HKU9-2	
23	EF065515	29136	Chiroptera	β-coronavirus	Bat coronavirus HKU9-3	
24	EF065516	29155	Chiroptera	β-coronavirus	Bat coronavirus HKU9-4	
25	EF424615	31017	Bos taurus	β-coronavirus	Bovine coronavirus E-AH65	
26	EF424616	30970	Bos taurus	β-coronavirus	Bovine coronavirus E-AH65-TC	
27	EF424617	31016	Bos taurus	β-coronavirus	Bovine coronavirus R-AH65	
28	EF424618	30995	Bos taurus	β-coronavirus	Bovine coronavirus R-AH65-TC	
29	EF424619	30995	Bos taurus	β-coronavirus	Bovine coronavirus E-AH187	
30	EF424620	30964	Bos taurus	β-coronavirus	Bovine coronavirus R-AH187	
31	EF424621	30995	Bovidae	β-coronavirus	Sable antelope coronavirus US/OH1/2003	
32	EF424622	30979	Giraffa camelopardalis	β-coronavirus	Giraffe coronavirus US/OH3-TC/2006	
33	EF424623	31002	Giraffa camelopardalis	β-coronavirus	Giraffe coronavirus US/OH3/2003	
34	EF424624	30979	Giraffa camelopardalis	β-coronavirus	Calf-giraffe coronavirus US/OH3/2006	
35	FJ376620	26476	Pycnonotus sinensis	δ -coronavirus	Bulbul coronavirus HKU11-796	
36	FJ415324	31029	Homo sapiens	β-coronavirus	Human enteric coronavirus 4408	
37	FJ425184	30995	Kobus ellipsiprymnus	β-coronavirus	Waterbuck coronavirus US/OH-WD358-TC/1994	
38	FJ425185	30940	Kobus ellipsiprymnus	β-coronavirus	Waterbuck coronavirus US/OH-WD358-GnC/1994	
39	FJ425186	30962	Kobus ellipsiprymnus	β-coronavirus	Waterbuck coronavirus US/OH-WD358/1994	
40	FJ425187	31020	Odocoileus virginianus	β-coronavirus	White-tailed deer coronavirus US/OH-WD470/1994	
41	FJ425188	30995	Rusa unicolor	β-coronavirus	Sambar deer coronavirus US/OH-WD388-TC/1994	
42	FJ425189	30997	Rusa unicolor	β-coronavirus	Sambar deer coronavirus US/OH-WD388/1994	
43	FJ647218	31285	Mus musculus	β-coronavirus	Murine coronavirus RA59/R13	
44	FJ647219	31427	Mus musculus	β-coronavirus	Murine coronavirus RJHM/A	
45	FJ647220	31429	Mus musculus	β-coronavirus	Murine coronavirus RA59/SJHM	
46	FJ647221	31456	Mus musculus	β-coronavirus	Murine coronavirus repA59/RJHM	
47	FJ647222	31283	Mus musculus	β-coronavirus	Murine coronavirus SA59/RJHM	
48	FJ647223	31386	Mus musculus	β-coronavirus	Murine coronavirus MHV-1	
49	FJ647224	31448	Mus musculus	β-coronavirus	Murine coronavirus MHV-3	
50	FJ647225	31293	Mus musculus	β-coronavirus	Murine coronavirus inf-MHV-A59	
51	FJ647226	31473	Mus musculus	β-coronavirus	Murine coronavirus MHV-JHM.IA	
52	FJ647227	31250	Mus musculus	β-coronavirus	Murine coronavirus repJHM/RA59	
53	FJ882963	29682	Homo sapiens	β-coronavirus	SARS coronavirus P2	
54	FJ884686	31275	Mus musculus	β-coronavirus	Murine hepatitis virus strain A59 B11 variant	
55	FJ938051	29232	Felis catus	α-coronavirus	Feline coronavirus RM	
56	FJ938052	29306	Felis catus	α-coronavirus	Feline coronavirus UU11	
57	FJ938053	29277	Felis catus	α-coronavirus	Feline coronavirus UU7	
58	FJ938054	29269	Felis catus	α-coronavirus	Feline coronavirus UU4	
59	FJ938055	29285	Felis catus	α-coronavirus	Feline coronavirus UU8	
60	FJ938056	29253	Felis catus	α-coronavirus	Feline coronavirus UU5	
61	FJ938057	29275	Felis catus	α-coronavirus	Feline coronavirus UU15	
62	FJ938058	28479	Felis catus	α-coronavirus	Feline coronavirus UU16	
63	FJ938059	29295	Felis catus	α-coronavirus	Feline coronavirus UU10	
64	FJ938060	29256	Felis catus	α-coronavirus	Feline coronavirus UU2	
65	FJ938061	29130	Felis catus	α-coronavirus	Feline coronavirus UU3	
66	FJ938062	29266	Felis catus	α-coronavirus	Feline coronavirus UU9	
67	FJ938063	31024	Bos taurus	β-coronavirus	Bovine coronavirus E-DB2-TC	
68	FJ938064	30995	Bos taurus	β-coronavirus	Bovine coronavirus E-AH187-TC	
69	FJ938065	30969	Bos taurus	β-coronavirus	Bovine respiratory coronavirus AH187	
70	FJ938066	30953	Bos taurus	β-coronavirus	Bovine respiratory coronavirus bovine/US/OH-440-TC/1996	
71	FJ938067	30953	Homo sapiens	β-coronavirus	Human enteric coronavirus strain 4408	
72	GQ153544	29695	NA	β-coronavirus	Bat SARS coronavirus HKU3-9	
73	GU553361	29264	Feliformia	α-coronavirus	Feline coronavirus UU22 isolate TCVSP-ROTTIER-00022	
74	GU553362	29264	Feliformia	α-coronavirus	Feline coronavirus UU23 isolate TCVSP-ROTTIER-00023	
75	HM211098	29136	Chiroptera	β-coronavirus	Bat coronavirus HKU9-5-1	
76	HM211099	29112	Chiroptera	β-coronavirus	Bat coronavirus HKU9-5-2	
77	HM211100	29136	Chiroptera	β-coronavirus	Bat coronavirus HKU9-10-1	
78	HM211101	29122	Chiroptera	β-coronavirus	Bat coronavirus HKU9-10-2	
79	HM245926	28915	Neovison vison	α-coronavirus	Mink coronavirus strain WD1133	
80	HQ392469	29233	Felis catus	α-coronavirus	Feline coronavirus UU40	
81	HQ392470	29255	Feliformia	α-coronavirus	Feline coronavirus UU19	
82	HQ392471	29252	Feliformia	α-coronavirus	Feline coronavirus UU20	
83	HQ392472	29233	Feliformia	α-coronavirus	Feline coronavirus UU30	
84	JF705860	27673	Anatidae	γ-coronavirus	Duck coronavirus isolate DK/CH/HN/ZZ2004	
85	JF792616	31286	Rattus	β-coronavirus	Rat coronavirus isolate 681	
86	JN183882	29243	Felis catus	α-coronavirus	Feline coronavirus UU47	
87	JN183883	29222	Felis catus	α-coronavirus	Feline coronavirus UU54	
88	JQ410000	27374	Vicugna pacos	α-coronavirus	Alpaca respiratory coronavirus isolate CA08-1/2008	
89	JQ989272	28483	Chiroptera	α-coronavirus	Hipposideros bat coronavirus HKU10 isolate TLC1343A	
90	JX860640	31028	Canis lupus familiaris	β-coronavirus	Canine respiratory coronavirus strain K37	
91	JX869059	30119	Homo sapiens	β-coronavirus	Human β-coronavirus 2c EMC/2012	
92	JX993987	29484	Rhinolophus pusillus	β-coronavirus	Bat coronavirus Rp/Shaanxi2011	
93	JX993988	29452	Chaerephon plicatus	β-coronavirus	Bat coronavirus Cp/Yunnan2011	
94	KC667074	30112	Homo sapiens	β-coronavirus	Human β-coronavirus 2c England-Qatar/2012	
95	KC776174	30030	Homo sapiens	β-coronavirus	Human β-coronavirus 2c Jordan-N3/2012	
96	KC881005	29787	Rhinolophus sinicus	β-coronavirus	Bat SARS-like coronavirus RsSHC014	
97	KC881006	29792	Rhinolophus sinicus	β-coronavirus	Bat SARS-like coronavirus Rs3367	
98	KF294457	29676	Rhinolophus monoceros	β-coronavirus	SARS-related bat coronavirus	
99	KF367457	30309	Rhinolophus sinicus	β-coronavirus	Bat SARS-like coronavirus WIV1	
100	KF569996	29805	Rhinolophus affinis	β-coronavirus	Rhinolophus affinis coronavirus isolate LYRa11	
101	KF906249	31052	Camelus bactrianus	β-coronavirus	Dromedary camel coronavirus HKU23 strain HKU23-265F	
102	KJ473811	29037	Rhinolophus ferrumequinum	β-coronavirus	BtRf-BetaCoV/JL2012	
103	KJ473812	29443	Rhinolophus ferrumequinum	β-coronavirus	BtRf-BetaCoV/HeB2013	
104	KJ473813	29461	Rhinolophus ferrumequinum	β-coronavirus	BtRf-BetaCoV/SX2013	
105	KJ473814	29658	Rhinolophus sinicus	β-coronavirus	BtRs-BetaCoV/HuB2013	
106	KJ473815	29161	Rhinolophus sinicus	β-coronavirus	BtRs-BetaCoV/GX2013	
107	KJ473816	29142	Rhinolophus sinicus	β-coronavirus	BtRs-BetaCoV/YN2013	
108	KJ473821	30423	Vespertilio sinensis	β-coronavirus	BtVs-BetaCoV/SC2013	
109	KJ481931	25406	Sus scrofa	δ -coronavirus	δ -coronavirus PDCoV/USA/Illinois121/2014 from USA	
110	KJ567050	25422	Sus scrofa	δ -coronavirus	Porcine δ -coronavirus 8734/USA-IA/2014	
111	KJ601777	25408	Sus scrofa	δ -coronavirus	δ -coronavirus PDCoV/USA/Illinois133/2014 from USA	
112	KJ601778	25404	Sus scrofa	δ -coronavirus	δ -coronavirus PDCoV/USA/Illinois134/2014 from USA	
113	KJ601779	25404	Sus scrofa	δ -coronavirus	δ -coronavirus PDCoV/USA/Illinois136/2014 from USA	
114	KJ601780	25404	Sus scrofa	δ -coronavirus	δ -coronavirus PDCoV/USA/Ohio137/2014 from USA	
115	KJ769231	25433	Sus scrofa	δ -coronavirus	Swine δ -coronavirus OhioCVM1/2014	
116	KM820765	25422	Sus scrofa	δ -coronavirus	Porcine δ -coronavirus KNU14-04	
117	KP886808	29723	Rhinolophus ferrumequinum	β-coronavirus	Bat SARS-like coronavirus YNLF_31C	
118	KT444582	30290	Rhinolophus sinicus	β-coronavirus	SARS-like coronavirus WIV16	
119	KU973692	29722	Chiroptera	β-coronavirus	UNVERIFIED: SARS-related coronavirus isolate F46	
120	KY352407	29274	Rhinolophus	β-coronavirus	Severe acute respiratory syndrome-related coronavirus strain BtKY72	
121	KY417142	29725	Aselliscus stoliczkanus	β-coronavirus	Bat SARS-like coronavirus isolate As6526	
122	KY417143	29741	Rhinolophus sinicus	β-coronavirus	Bat SARS-like coronavirus isolate Rs4081	
123	KY417150	30311	Rhinolophus sinicus	β-coronavirus	Bat SARS-like coronavirus isolate Rs4874	
124	KY417151	30307	Rhinolophus sinicus	β-coronavirus	Bat SARS-like coronavirus isolate Rs7327	
125	MG772933	29802	Rhinolophus sinicus	β-coronavirus	Bat SARS-like coronavirus isolate bat-SL-CoVZC45	
126	MG772934	29732	Rhinolophus sinicus	β-coronavirus	Bat SARS-like coronavirus isolate bat-SL-CoVZXC21	
127	MK211374	29648	Rhinolophus	β-coronavirus	Coronavirus BtRl-BetaCoV/SC2018	
128	MK211375	29698	Rhinolophus affinis	β-coronavirus	Coronavirus BtRs-BetaCoV/YN2018A	
129	MK211376	30256	Rhinolophus affinis	β-coronavirus	Coronavirus BtRs-BetaCoV/YN2018B	
130	MK211377	29689	Rhinolophus affinis	β-coronavirus	Coronavirus BtRs-BetaCoV/YN2018C	
131	MK211378	30213	Rhinolophus affinis	β-coronavirus	Coronavirus BtRs-BetaCoV/YN2018D	
132	MN908947	29903	Homo sapiens	β-coronavirus	Severe acute respiratory syndrome coronavirus 2 (Wuhan)	
133	MN996532	29855	Rhinolophus affinis	β-coronavirus	Bat coronavirus RaTG13	
134	MT121216	29521	Manis javanica	β-coronavirus	Pangolin coronavirus isolate MP789	
135	NC_006577	29926	Homo sapiens	β-coronavirus	Human coronavirus HKU1	
136	NC_009019	30286	Chiroptera	β-coronavirus	Bat coronavirus HKU4-1	
137	NC_009020	30482	Chiroptera	β-coronavirus	Bat coronavirus HKU5-1	
138	NC_009021	29114	Chiroptera	β-coronavirus	Bat coronavirus HKU9-1	
139	NC_010646	31686	Delphinapterus leucas	γ-coronavirus	Beluga Whale coronavirus SW1	
140	NC_010800	27657	Meleagris gallopavo	γ-coronavirus	Turkey coronavirus	
141	NC_011547	26487	Pycnonotus jocosus	δ -coronavirus	Bulbul coronavirus HKU11-934	
142	NC_011549	26396	Turdus hortulorum	δ -coronavirus	Thrush coronavirus HKU12-600	
143	NC_011550	26552	Lonchura striata	δ -coronavirus	Munia coronavirus HKU13-3514	
144	NC_012936	31250	Rattus	β-coronavirus	Rat coronavirus Parker	
145	NC_014470	29276	Rhinolophus blasii	Unknown	Bat coronavirus BM48-31/BGR/2008	
146	NC_016992	26083	Passeridae	δ -coronavirus	Sparrow coronavirus HKU17	
147	NC_016993	26689	Muscicapidae	δ -coronavirus	Magpie-robin coronavirus HKU18	
148	NC_016994	26077	Ardeidae	δ -coronavirus	Night-heron coronavirus HKU19	
149	NC_016995	26227	Mareca	δ -coronavirus	Wigeon coronavirus HKU20	
150	NC_016996	26223	Gallinula chloropus	δ -coronavirus	Common-moorhen coronavirus HKU21	
151	NC_017083	31100	Oryctolagus cuniculus	β-coronavirus	Rabbit coronavirus HKU14	
152	NC_018871	28494	Chiroptera	α-coronavirus	Rousettus bat coronavirus HKU10	
153	NC_019843	30119	Homo sapiens	β-coronavirus	Middle East respiratory syndrome coronavirus	
154	NC_023760	28941	Neovison vison	α-coronavirus	Mink coronavirus strain WD1127	
155	NC_025217	31491	Hipposideros pratti	β-coronavirus	Bat Hp- β-coronavirus/Zhejiang2013	
156	NC_026011	31249	Rattus norvegicus	β-coronavirus	β-coronavirus HKU24 strain HKU24-R05005I	
157	NC_028752	27395	Camelus	α-coronavirus	Camel α-coronavirus isolate camel/Riyadh/Ry141/2015	
158	NC_028806	28111	Sus scrofa	α-coronavirus	Swine enteric coronavirus strain Italy/213306/2009	
159	NC_028811	27935	Myotis ricketti	α-coronavirus	BtMr-AlphaCoV/SAX2011	
160	NC_028814	27608	Rhinolophus ferrumequinum	α-coronavirus	BtRf-AlphaCoV/HuB2013	
161	NC_028824	26975	Rhinolophus ferrumequinum	α-coronavirus	BtRf-AlphaCoV/YN2012	
162	NC_028833	27783	Nyctalus velutinus	α-coronavirus	BtNv-AlphaCoV/SC2013	
163	NC_030292	28434	Mustela putorius	α-coronavirus	Ferret coronavirus isolate FRCoV-NL-2010	
164	NC_030886	30161	Rousettus leschenaultii	β-coronavirus	Rousettus bat coronavirus isolate GCCDC1 356	
165	NC_032107	28363	Triaenops afer	α-coronavirus	NL63-related bat coronavirus strain BtKYNL63-9a	
166	NC_032730	28763	Rattus norvegicus	α-coronavirus	Lucheng Rn rat coronavirus	
167	NC_034440	29642	Pipistrellus	Unknown	Bat coronavirus isolate PREDICT/PDF-2180	
168	NC_034972	27682	Apodemus chevrieri	α-coronavirus	Coronavirus AcCoV-JC34	
169	NC_035191	25995	Suncus murinus	α-coronavirus	Wencheng Sm shrew coronavirus	
170	NC_038294	30111	Homo sapiens	β-coronavirus	β-coronavirus England 1	
171	NC_038861	28586	Sus scrofa	α-coronavirus	Transmissible gastroenteritis virus genomic RNA	
172	NC_039207	30148	Erinaceus europaeus	β-coronavirus	β-coronavirus Erinaceus/VMC/DEU/2012	
173	NC_039208	25425	Sus scrofa	δ -coronavirus	Porcine coronavirus HKU15 strain HKU15-155	

For each type of structural protein, a matrix of distance between amino acid sequences was normalized to the maximum from the combined perspective of rows and columns. The normalized score of each cell depends on the respective row and column maxima simultaneously (mean fraction). So, final values fall symmetrically within the unit interval [0, 1]. The correlation between matrices of distances was studied via Mantel’s test. A single matrix of distances between viral units was obtained as a linear combination of the normalized distances between involved proteins. We construct a proximity network from such a unified matrix of viral distance. This proximity network refers to the minimum spanning tree that connects all sampled viruses at the minimum cost (i.e., the sum of distances across edges is minimized). The proximity network encodes the backbone of the similarity relationships between studied items.

After identifying the set of hosts defined to the highest possible degree of taxonomic resolution, we focused on the phylogenetic tree subtended by them. The phylogeny for the vertebrate hosts was recovered from the VertLife dataset at http://vertlife.org (mammals: Upham, Esselstyn & Jetz, 2019; birds: Jetz et al., 2012). Branch lengths of the tree were computed using the method of Grafen (1989). The distance to root is set to 1. The patristic distance was adopted as a measure of phylogenetic distance between terminals, namely the sum of the lengths of the branches that link two taxa at the leaves of the tree. The phylogenetic diversity associated with a set of taxa was derived from the average value of pairwise patristic distances. Then, we match any available host with the pertinent phylogenetic node, either a terminal or an inner node, depending on the taxonomic resolution of the item. This step is necessary for calculating later the phylogenetic distance between hosts. Computational null experiments were run to assess the coupled information between the phylogeny of hosts and the configuration of similarities between viral entities.

Chimerism was theoretically studied. Structural proteins of the same kind are here called homotopic (e.g., orthologous pairs Sx − Sy, Ex-Ey, Nx-Ny, Mx-My coming from viral sources x and y), otherwise they are called heterotopic (non-orthologous pairs Sx-Ey; Sx-Ny; Sy-Mx and all possible cross combinations). We can establish the distances among homotopic proteins, but this cannot be done among heterotopic ones. Therefore, we used the distances among homotopic proteins to infer the associations between the heterotopic ones, based on the crossed distances they have with the respective homotopic proteins. The inferred associations between the heterotopic proteins are defined as heterotopic disaffinity.

Whenever high correlation among matrices of distances is detected, the next assumption would gain support: Similar homotopic proteins (low distance between them) are likely exchangeable in the assembly they occur. So, for two heterotopic proteins recorded in different virions, their feasibility of being combined into a new theoretical structure can be estimated from the similarity between homotopic elements of the virions where they are actually embedded into. We will refer to this as the interchangeability property of structural proteins (IPSP). The lower the heterotopic disaffinity between a pair of proteins, the larger the chance of being integrated into a common viral assemblage. As a corollary of this statement, chimerism understood as a mosaic of proteins already recognized in distant viruses, is hardly expected to occur in nature. The average of pairwise disaffinities in a tetrameric assembly estimates its degree of chimerality. Using combinatorial simulations, we study the behavior of the coefficient of chimerality in mixtures of proteins randomly drawn from the empirical sets already compiled by us.

Finally, we assess the co-structure between phylogeny and chimerism in our dataset. We run computational experiments of chimeras (combinatorial urn models) that represent our theoretical morphospace and we study their association with the content of phylogenetic information. We draw 100,000 proteins of classes E, M, N, and S from the respective urns. Then, we assemble them in tetrads. In parallel, we calculate the phylogenetic diversity associated with the pool of hosts in which the sampled proteins were observed to occur. All statistical tests, analyses, and graphics were carried out with the R software (R Core Team, 2020, version 4.0.0). See supplementary material: https://github.com/GFontanarrosa/Viral-Morphospace-Dos-Santos-et-al/blob/main/Coronavirinae_complete_analysis.R.

Results

Basic information about amino acid sequences of the major structural proteins (E, M, N, and S) is reported. Proteins can be strictly ordered by length, i.e., E < M < N < S, across the entire sample of sequences (Table 2). A significant correlation was detected among the four matrices of distances (P < 0.01, Mantel’s test), suggesting that they are congruent regarding the configuration of pairwise similarities between data points. Correlation scores are consistently higher than 0.7 (Table 3). This finding suggests the average distance between proteins serves as a proxy to assess the dissimilarity between virions as a whole.

Table 2 Size of coronavirus structural proteins (length of sequence).

The Tukey Five-Number Summaries are the maximum and minimum values, the lower and upper quartiles, and the median of the data set.

	E protein	M protein	N protein	S protein	
Tukey Five-Number Summaries	[65, 76, 82, 83, 109]	[185, 221, 222, 230, 268]	[342, 379, 421, 448, 470]	[1126, 1241, 1324, 1363, 1472]	
Mean (SD)	80.7 (4.8)	229.9 (15.8)	413.5 (35.5)	1308.7 (91.2)	

Table 3 Structural protein distances.

Correlation between matrices of distances. All values are statistically significant after performing Mantel’s test (P < 0.01).

	E	M	N	S	
E	–	0.83	0.82	0.74	
M	–	–	0.87	0.80	
N	–	–	–	0.79	
S	–	–	–	–	

Figure 1 depicts the minimum spanning tree (proximity network) of Coronavirinae isolates. Topologically, the four genera of Coronavirinae could be segregated into connected components after removing the only inter-genera links. The α-CoVs lie always at the intermediacy along the shortest paths connecting β-CoVs with the remaining coronaviruses genera. The two unknown items (nodes 167 and 145) are located within the β-coronavirus set. SARS-CoV-2 is located at just one-step distance from the network periphery and lies between the RaTG13 (the peripheral node 133) and the pangolin-CoV-2020 (the inner node 134) (Fig. 1). In terms of protein distances, SARS-CoV-2 is consistently closer to RaTG13 than any other sequenced element. Additional information about nodes is displayed in Table 1.

Figure 1 Proximity network spanning over 173 samples from the group Coronavirinae of viruses.

The four distinct CoV genera can be easily segregated after removing the unique between-genera links, and are highlighted through a gray halo. Nodes have been colored by clade membership of host in which virus was isolated. SARS-CoV-2 and adjacent nodes have been tagged with the respective host icon. Human silhouette was also added to all those viruses infecting humans. Note the overall co-structure between viral proteome distance and phylogenetic distance of respective hosts, leading to a broad agreement between connected clusters of CoV genera and host clades. Additional information about nodes of the network are available in Table 1. Silhouette images were freely obtained from http://phylopic.org/.

The minimal set of links of MST grasps the skeleton of relationships between viral samples. It synthesizes the structure of similarities held by data. Focusing on it, the main patterns are easily recognized and hypothesis generation becomes facilitated. In observing the network links of Fig. 1 with nodes tagged with the respective host, we track the phylogenetic relatedness between pairs of hosts across the total set of links (Fig. 2). In Fig. 2, each link is represented like a parabolic arc between hosts at the terminals in the phylogeny. The height of the parabola is dictated by the distance between nodes of Fig.1. A minor proportion of links (14%) bridge less similar viruses (distance > 0.1) and are frequently associated with weakly related hosts (0.65 ± 0.13, mean ± standard error of patristic distance). The staircase pattern of parabolic arcs shown by birds is eloquent in this regard (Fig. 2). We test this claim through randomization. We run 10,000 random experiments of host allocation in the same network or backbone of proximity relationships. Phylogenetic distance between neighboring hosts increases steeply in the random scenarios (Fig. 3). The observed distribution of hosts across the proximity network is compact in phylogenetic terms. In general, hosts of very similar viruses are also close phylogenetically.

Figure 2 Graphical representation of hosts associated with the endpoints of links in the proximity network of Fig. 1.

To the left, phylogenetic tree of involved hosts. To the right, links/edges of proximity network represented as parabolic arcs bridging the hosts associated with endpoints of such links. The height of arcs correspond to the distance between nodes/virus connected by the respective link, so that flat arcs represent links between similar viruses whereas bumpy arcs join dissimilar ones. All taxa from the main clades (highlighted through transparent rectangles) retrieve always a between-clade patristic distance larger than unity (>1.0). Silhouette images were freely obtained from http://phylopic.org/.

Figure 3 Expected phylogenetic distance between hosts under random scenarios of host allocation on the same proximity network represented in Fig. 1.

Quantiles of viral distance are plotted against quantiles of phylogenetic distance between hosts. Dotted polyline, the observed distribution of values. Solid polyline, values obtained after randomization. The 95% confidence interval is drawn around this last line. Departure of observed values from randomness indicates that hosts of viruses directly connected in the proximity network tends to be closely related.

Figures 4A–4B depict, in a didactic way, the flow work leading to the calculation of chimerality. Figure 4A shows three hypothetical viral configurations. The heterotopic disaffinity is calculated from the distance between involved homotopic proteins. Thus, for instance, the heterotopic disaffinity between protein M from the leftmost viral configuration and the s protein from the rightmost one comes from the mean distance between the respective homotopic partners (i.e., distances M-m and S-s). Figure 4B shows the 81 possible tetrads (hypothetical theoretical morphospace) obtained by a free combination of protein precursors. This figure highlights both the heterotopic disaffinity between pairs of proteins of each configuration and the chimerality coefficient of the whole configuration.

Figure 4 Computational experiments of chimera compositions.

Didactic introduction to concepts (A–B) in addition to results from such experiments (C) applied on our real data. (A) Three hypothetical tetrads of structural proteins coming from three different viruses. The distance between them are indicated (normalized values to the maximum between brackets). Here, the distance denotes the amount of differences in the attributes of letters used to label the protein (upper/lowercase; normal/italics). (B) Showing all the possible combinations of proteins from the above hypothetical viral sources. Heterotopic disaffinity between pairs of distinct proteins is inferred from the distance between proteins of the same kind of the viral precursors. For any assembly, the degree of chimerality is the average heterotopic disaffinity.

Results of computational experiments of chimeras are plotted in Fig. 4C. It shows the dispersion of both chimerality and phylogenetic diversity of hosts in the random set of tetrads. The frequency of observations is represented through a heatmap. Noticeably, the rarest event is to find simulated viruses that jointly exhibit high intrinsic phylogenetic diversity and low coefficient of chimerality. Since different viruses can be recognized in closely related hosts, it is possible to achieve tetrads of high chimerism (low phylogenetic diversity, high coefficient of chimerality). On the contrary, it is rather difficult to find similar viruses in loosely related organisms (high phylogenetic diversity, low coefficient of chimerality).

Discussion

Our analysis of structural proteins recovered both the four viral genera (α, β, γ, and δ) and SARS-CoV-2 affinities with viruses isolated from bats and pangolins. The approach is useful to address issues of taxonomic classification such as positioning of unknown items. Rapid classification of new viruses is a topic of great concern since it contributes for strategic planning, containment, and treatment (Randhawa et al., 2020). In the proximity network, the β-coronavirus and α-coronavirus sets are neighbors. The β-coronavirus set is structured into two subgroups that are bridged by a sequence of the nodes 95-91-153-170, all isolated from Homo sapiens. Almost all the viruses that infect human hosts in our sample are distributed in the left subgroup regarding node 170, with the exception of SARS-CoV-2 (node 132) and the SarsCovP2 (node 53) which are located in the right subgroup. Considering the remarkable proteomic closeness among most viruses infecting humans (Fig. 1), it could be inferred that viruses located in the vicinity of SARS-CoV-2 are also potentially dangerous to humans. The fact that SARS-CoV-2 rather than their neighbors in the proximity network has emerged recently in the human population could be due to the degree of biogeographic and ecological isolation of its hosts or lack of opportunity (Segreto & Deigin, 2020; Solé & Elena, 2018). The limit between α-coronaviruses and β-coronaviruses is depicted by node 135, also a human parasite. Viruses historically infecting a wide range of vertebrate hosts seem to be converging to infect humans. Human explosive demography jointly with human-driven changes as bringing in close contact farm animals and crops with wild animals and plants are the triggers of viral evolution and spillovers (Woolhouse, Taylor & Haydon, 2001). Notwithstanding, considerations about the bridging role of humans in diversification of β-coronavirus should be taken with caution because of biasing in datasets (e.g., NCBI Virus) towards viral sequence from isolates infecting humans.

The RaTG13 (node 133 isolated from Rhinolophus affinis), previously identified as the closest known relative of SARS-CoV-2 based on genome similarity (Cyranoski, 2020; Zhang & Holmes, 2020; Zhang et al., 2020), is located peripherally in the β-coronavirus set and is the immediate neighbor of the SARS-CoV-2. The two unknown items presumably belong to the genus β-coronavirus based on the membership of their local neighborhood in the network (Fig. 1). Node 134 (isolated from Manis javanica) is also connected with SARS-CoV-2 but showing an inner location within the network. The subset composed of RaTG13, SARS-CoV-2, and pangolin-CoV-2020 is in turn located peripherally in the main network. The peripheral position of RatG13 may be related to its isolated evolution in the Yunnan’s caves (Southern China) where R. affinis inhabits. Accessibility to these caves for researchers did not occur until recently (Segreto & Deigin, 2020).

Coronaviruses infect a range of mammalian and avian species (Latinne, Hu & Olival, 2020). Within them, α-coronaviruses are able to switch hosts more frequently and between more distantly related taxa than β-coronaviruses. These last are specialist strategists infecting mainly bats and also other mammalian species such as humans, camelids, and leporids (Figs. 1 and 2). Nevertheless, the emergence of SARS-CoV-2 suggests a jump between phylogenetically distant hosts, allowed by modifications in the RBD that make it more virulent and host-specific for humans. This modification enables a new range of potential hosts for SARS-CoV-2 (hosts phylogenetically related to humans and domestic and farm animals that co-inhabit with humans).

Our results showed that more similar viruses tend to infect the most phylogenetically related hosts displaying a specialist strategy (Fig. 2). This result is reinforced by the randomized simulations here performed (Fig. 3). Longdon et al. (2011) found evidence that most host shifts occur between closely related hosts, and that the host phylogeny could explain most of the variation in viral replication and persistence. Viruses that co-evolved with a certain species of vertebrates have developed host-specific mechanisms to infect it. This adaptation will be more likely co-opted as an exaptation to jump into a host species closely related to the host in which the virus evolved (Latinne, Hu & Olival, 2020). This specialist strategy is held by the majority of viruses (Solé & Elena, 2018) and represents an ecological constraint on the virus-host available set. However, there are also viruses separated by long distances infecting closely related hosts such as Mareca sp. and Meleagris sp. In this case, the distance is of 0.53 between viruses and belong to δ- and γ- genera, respectively. On the contrary, there are a few viruses with shorter distances infecting distantly related hosts, as in the case of Homo sapiens and Rhinolophus affinis. Succinctly, results show: (1) The minimum spanning network recapitulates the known phylogeny of Coronovirinae, and (2) some concordance is found between host phylogeny and viral genetic distance. With a few exceptions, this result suggests that the overall pattern is not one of frequent host shifts.

Since bats are natural reservoirs for several coronaviruses that can potentially infect humans (Woo et al., 2012), their viruses have been deeply studied and even researchers have been using them to generate chimera coronaviruses for the last 20 years (Segreto & Deigin, 2020). Laboratory chimeras were meant to simulate recombination events that might occur in nature (Menachery et al., 2015). Thus, even when a chimera virus is detected, the distinction between natural and artificial chimeras represent another challenging step. We use the IPSP to obtain the different possibilities of theoretical viruses and relate them to the degree of chimerality. The larger the amount of interactions between proteins coming from dissimilar virus sources, the larger the chimerality of that particular assemblage in the sense of decreasing chance for observing it in nature (lower feasibility). Our results on chimeral virus simulations (Fig. 4C) showed a non-trivial fill of the theoretical morphospace. Whenever protein precursors come from phylogenetically distant hosts, chimerality is expected to achieve high values in the sense of a global entity composed of dissimilar, heterogeneous parts. The relevance of this approach is that it gives us clues to assess the chimeral origin of coronaviruses. To inquire about a potential chimera origin of a certain sampled virus, we can compare it with the viruses belonging to the empirical morphospace and the theoretical morphospace. If our focal virus turns out to be more similar to a theoretical chimera virus (belonging to the theoretical morphospace) than to an observed one (empirical), then the suspicion about a chimera origin increases.

Based on the aforementioned insights, our analysis cannot support hypothesis number 2, since SARS-CoV-2 does not have all the features deemed to be chimeric using just information about amino acid sequences. Even though the chimerism origin theory is consistent with the remarkable proteomic closeness between RaTG13, SARS-CoV-2 and pangolin-CoV-2020, also found in this work, we also observe that when extracting the SARS-CoV-2 from Fig. 1, pangolin-CoV-2020 and RaTG13 do not undergo modifications in their network locations. Andersen et al. (2020) also state that the genetic data irrefutably show that SARS-CoV-2 is not derived from any previously backbone used in chimeras assemblage. Our approach represents a tool that could guide researchers to detect chimerality.

After phylogenetic genome-wide analysis, most studies indicated that Rhinolophus bats may be the natural host of the novel coronavirus (Ma et al., 2021). However, these results should be critically appraised since classic dichotomic phylogenetic tools do not handle recombination well and results could be misleading if recombination occurs (Goh et al., 2022; Posada, 2000). In order to deal with these constraints, Goh et al. (2022) suggest narrowing the phylogenetic analysis to conserved proteins such as the M protein. In doing so, a different tale emerges and pangolin is no longer so easily dismissed as ancestor. We also consider that network and combinatorial approaches can be useful to address issues of recombination. Thus, the intermediary position of SARS-CoV-2 between pangolin and bat calls for a care consideration. New lineages as a result of blending of loosely related predecessors, for instance symbionts or hybrids, pose a challenge for classic phylogenetic reconstruction. Alternatively, phylogenetic networks allow investigation of complex evolutionary histories that involve cross-species gene transfer (Albrecht et al., 2012). On the other hand, combinatorics under a morphospace research program shed light on how likely an entity can occur in nature. Our proposal then expands the repertoire of biocomputational resources to gain a deeper understanding of evolution of items through events other than cladogenesis or speciation.

Conclusions

Since WHO declared the COVID-19 outbreak a pandemic on March 11, 2020, an unprecedented multidisciplinary interest in the responsible coronavirus exploded from all around the world at the same time. One approach to gain a better comprehension of it is by zooming in their structural details. Another approach consists of zooming out and achieving the big picture of Coronavirinae as a whole. We provide a general framework to address issues of viral classification, assembly constraints, degree of chimerism, evolutionary paths, and putative chains of zoonotic jumps. We constructed a proximity network based on the four major structural proteins in coronavirus. Through this, we explored the relationship between host-phylogeny and viral proteomic distance. We also investigated the potential of generating feasible chimeras in nature from loosely related hosts through simulation. Finally, we brought attention to both the molecular and phylogenetic constraints behind the evolution of coronaviruses.

We would like to thank all the people in the academic and civil world in general who, through their effort and sacrifice, have managed to keep us informed and safe during the pandemic.

Additional Information and Declarations

Competing Interests

Author Contributions

Data Availability

Virginia Abdala is Academic Editor for PeerJ.

Daniel Andrés Dos Santos conceived and designed the experiments, performed the experiments, analyzed the data, prepared figures and/or tables, authored or reviewed drafts of the article, and approved the final draft.

María Celina Reynaga conceived and designed the experiments, analyzed the data, prepared figures and/or tables, authored or reviewed drafts of the article, and approved the final draft.

Juan Cruz González performed the experiments, analyzed the data, authored or reviewed drafts of the article, and approved the final draft.

Gabriela Fontanarrosa analyzed the data, prepared figures and/or tables, authored or reviewed drafts of the article, and approved the final draft.

María de Lourdes Gultemirian analyzed the data, authored or reviewed drafts of the article, and approved the final draft.

Agustina Novillo analyzed the data, authored or reviewed drafts of the article, and approved the final draft.

Virginia Abdala analyzed the data, authored or reviewed drafts of the article, and approved the final draft.

The following information was supplied regarding data availability:

The raw data is available at GitHub: https://github.com/GFontanarrosa/Viral-Morphospace-Dos-Santos-et-al/blob/main/Coronavirinae_complete_analysis.R.

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
