# Peer review of "Insights on the evolution of Coronavirinae in general, and SARS-CoV-2 in particular, through innovative biocomputational resources"

_PeerJ, doi:10.7717/peerj.13700_

## Round 0.1 · original submission · Major Revisions

Reviewer 2 has raised that your reporting of the methodology is not sufficiently detailed for them to be able to replicate your analysis. Please provide further details as well as addressing the other points raised by both reviewers.

Reviewer 1 ·

Basic reporting

Figure 1 could be improved if some animal hist notations could be placed next to some of the nodes of the tree, In this way,, evolution of CoVs among its animal hosts can be observed.

Experimental design

This is an interesting paper that introduces a novel methodology to phylogenetic studies.
The authors did, however, applied to just one of the many problems involving the use of phylogenetic studies in COVID-19 related. The paper attempts to address the problem pf chimerism especially with respect to the possible lab design of the virus. While this is a important problem, there are, however, many other unsolved problems.

A glaring example is the search for the intermediary animal host. Pangolin and bats have been cited as possible intermediaries. Many are not convinced that pangolins are the intermeidary for two reasons. Firstly, the sequence similarity between pangolin-cov and SARS-CoV-2 is about 90%, which some people believe to be insufficiently close. Secondly, there are much confusion about the phylogenetics:
https://pubmed.ncbi.nlm.nih.gov/32407364/
As seen in the article above, the phylogenetic study using genome-wide is unable to conclude that SARS-CoV-2 came from pangolin-cov. Goh et al suggested that much of the confusion arose because the existing phylogenetic algorithms is not able to handle recombination well:
https://pubs.acs.org/doi/10.1021/acs.jproteome.2c00001
https://pubmed.ncbi.nlm.nih.gov/11114432/
They suggested that the best phylogenetic tree can be obtained by using only the M protein as the M protein is highly conserved among all COVID-19 related viruses because the COVID-19 M protein is among the hardest within the CoV family. Indeed, the COVID-19 phylogenetic study using M protein shows a tree that is different from those using other proteins or entire genome:
https://zenodo.org/record/6355263#.YkYbvh0RXUr
I was looking to see if there is any resulting difference between the approach used by this paper and the studies already done especially with respect to this issue. I am not able to see the difference as the paper did not include sufficient samples of COVID-19 CoVs. It would have made the paper stronger if the authors are able to demonstrate the differences.

Validity of the findings

The approach is novel interesting and perhaps good but the authors need to make their case stronger as mentioned above. The paper needs to show if it can handle recombination better than current phylogenetic software available.

Reviewer 2 ·

Basic reporting

The authors constructed a proximity network based on the 4 major structural proteins in coronavirus. Through this, the authors explored the relationship between host-phylogeny and viral proteomic distance. They also investigated the potential of generating feasible chimeras in nature from loosely related hosts through simulation.

Experimental design

The concept of the study is novel and interesting. Given the methods used in the study is atypical, a clear explanation of the methods is paramount. Unfortunately, it is not well written, which made following the methods, let along replicating it, particularly difficult. I cannot confidently give a more in depth review or feedback.

Validity of the findings

The methodology sections of the study needs to be better written. I am unable to replicate the study as it is.

Additional comments

Line 130-139. I appreciate the authors have in depth discussion on how this analysis was calculated, but it is not clear to me and I would not be able to replicate this. A few terminologies are unclear to me, such as “pertinent phylogenetic node”. The sentence - “the phylogenetic diversity associated with a set of taxa was derived from the average value of pairwise patristic distances” – what does “a set of taxa” refer to? “Match any available host with the pertinent phylogenetic node, either a terminal or an inner node” – are the nodes referring to the viruses? How was that used to calculate the phylogenetic distance between hosts (isn’t that calculated using patristic distance mentioned in line 131)?
Line 150. “So, for two heterotopic proteins, the chance of being amalgamated increases if both are similar to the respective partners in the actual assembly where the alter occurs.” This sentence does not make sense to me.
Line 154. “As a corollary of this statement, chimeras derived from the assembly of proteins coming from slightly akin viruses hardly proceed in a natural way.” Not sure what the sentence means.
Line 157. “Using combinatorial simulations, we study the behavior of the coefficient of chimerality in mixtures of proteins randomly drawn from the empirical sets already compiled by us.” As in compiled in the earlier section of the method? Or in-house pre-compiled set? If the latter, how was the compiling done?
Line 171. “Basic information about amino acid sequences of every kind of protein” sounds a little informal – perhaps use terms like “Basic information about amino acid sequences of the major structural proteins”
Line 172. “Lengths of sequences vary across ranges that never overlap when comparing among 173 them (Table 2)”. I am not sure what the sentence mean. The length of sequence are not identical in any of the 4 proteins for any of the 173 genomes?
Line 175. “suggesting that the information held by the structural proteins is rather concordant.” I am guessing what the authors meant by information, they meant “cross-genome variation in structural proteins”, but it is not clear and the sentence does not really make much sense as it is (what is “the information” held by structural protein?).
Line 176. “This finding makes the average distance between proteins a good choice to assess the dissimilarity between virions as a whole.” I agree that it is a good proxy, but more rigorously analysis, such as whether if the variation occur in conserved regions, would be required to claim this to be a good choice. Also, I would change the language from “makes the average distance…” to “suggests the average distance…”.
Line 179. “perfect connected components after removing the only inter-genera links.” It is unclear to me what perfect connected component means.
Line 180. “All the shortest paths between β-coronavirus and γ- and δ-coronavirus necessarily pass through representatives of α-coronavirus.” Please rewrite the sentence.
Line 183. “SARS-CoV-2 is located at the frontier of the network” What do you mean by “frontier of the network”? It’s not a terminal node, is frontier a special terminology used in network analysis?
Line 187. I am not entirely sure I follow the reasoning for Figure2. Am I correct in thinking that the authors are attempting to show that two nodes with a link between are more likely to bridge more similar virus than less similar virus? But isn’t that the whole point of the network? The majority of the edges would be between viruses within the same genera, with the occasional cross-genera edge that will have large parabolic arcs.
Line 187. “By focusing exclusively on the network links of Figure 1, and tagging nodes with the respective host, we track the phylogenetic relatedness between pairs of hosts across the total set of links (Fig. 2).” It is not clear to me why relatedness between pairs of hosts across total set of links were only done on the network links in Figure 1 rather than an all-to-all comparison. If there is a reason for this, please clarify this in the text.
Line 189-193. Please be more specific (e.g. include the percentages or definition) with the terms “minor proportion”, “frequently associated”, “weakly related hosts”.
Line 191. “Viral endpoints of links”. I am unsure what this refers to.
Figure 1. It may help readers to explore potential relationships of host organisms within the network if the nodes were outlined by a second colour indicating the host organism. Can edge thickness be used to indicate the distance (such as thickness correlates to the invert log distance) between the connected nodes?
Figure 1. Are the two grey circles with question marks the two unknowns? If so, please indicate in figure legend.

---

## Round 0.2 · accepted · Accept

Thank you for addressing the reviewers' critiques.

Reviewer 1 ·

Basic reporting

All requirements met as far as I can see. Improvements made since last review

Experimental design

Improvements made since last review

Validity of the findings

Improvements seen since last review